# Personality Determinants of Diet Health Quality among an Elite Group of Polish Team Athletes

**DOI:** 10.3390/ijerph192416598

**Published:** 2022-12-10

**Authors:** Maria Gacek, Agnieszka Wojtowicz, Adam Popek

**Affiliations:** 1Department of Sports Medicine and Human Nutrition, Faculty of Biomedical Sciences, University of Physical Education in Kraków, 31-571 Krakow, Poland; 2Department of Psychology, Faculty of Social Sciences, University of Physical Education in Kraków, 31-571 Krakow, Poland; 3Bronisław Markiewicz State Higher School of Technology and Economics in Jarosław, 37-500 Jarosław, Poland

**Keywords:** personality traits, the big five five-factor model, diet health quality indices, sports nutrition, team sports

## Abstract

The aim of the study was to analyse personality determinants of diet health quality among of an elite group of Polish team athletes. The correlations between personality traits of the big five model and the indices of diet health quality (the pro-healthy diet index, pHDI-10 and the non-healthy diet index, nHDI-14) were assessed. Research was carried out among 213 athletes (males), using the beliefs and eating habits questionnaire (Kom-PAN) and the neuroticism extraversion openness personality inventory—revised (NEO-PI-R). Statistical analysis was performed with the use of Pearson’s linear and Spearman’s signed rank correlation coefficients as well as multiple regression, assuming the significance level of α = 0.05. It has been shown that the pro-healthy diet index (pHDI-10) decreased with increasing *Neuroticism*, while an increase was noted along with increasing *Extraversion*. In the case of the non-healthy diet index (nHDI-14) a decrease along with a simultaneous increase in the level of *Agreeableness* (*p* < 0.05). Significant (but weak) correlations have been indicated between personality traits and diet health quality. The identified dependencies may be used to personalise the impact of dietary education among athletes.

## 1. Introduction

The achievement of sports goals is conditioned not only by a wide range of motor and mental abilities [1,2,3], but also by quality of diet, considering the increased demand for energy, fluids and some nutrients (carbohydrates, proteins, B vitamins, antioxidants and mineral salts). A varied diet, rich in products with high nutritional density and limiting those with low nutritional density, is one of the key factors contributing to health, improving exercise capacity and the effective post-exercise recovery of athletes [4,5].

One of the proposals for the rational nutrition of athletes is the model of the Swiss pyramid, which opens with water and other non-sweetened drinks, and ends with sweet and salty snacks. The intermediate levels of the pyramid are occupied by: vegetables and fruits, wholegrain cereal products, legumes, other protein products and vegetable fats, recommended for consumption in various amounts, and at a certain frequency. This model emphasizes the special importance of water for the regulation of hydration and electrolyte management, as well as vegetables and fruits for maintaining appropriate antioxidant status and acid-base balance in conditions of vigorous physical exercise [6].

The health quality of a diet, related to the implementation of quantitative and qualitative dietary recommendations, can be assessed using various indices, including the healthy diet index (HEI), the Australian diet quality index (Aussie-DQI) [7,8,9] and the pro-healthy diet (pHDI-10) and non-healthy diet (nHDI-14) indices functioning in the Polish literature. The pro-healthy diet index concerns the frequency of consuming products with potentially beneficial effects on health, while the non-healthy diet index concerns the consumption of products potentially detrimental to health [10].

Nutritional behaviour is determined by numerous factors, including environmental and individual conditions [11,12,13]. An important area of determinants regarding food choices are psychological factors, including personality traits. Personality is treated as an internal regulatory system that allows for adaptation to selected situations and the environment as well as internal integration of thoughts, feelings and behaviours [14]. One of the models regarding the approach to personality and its description is the five-factor model of personality by Costa and McCrae, which, in the last few decades, has become one of the dominant paradigms in the psychology of traits [15]. The so-called big five model by Costa and McCrae [16] consists of five main dimensions of personality, including *Neuroticism*, *Extraversion*, *Openness to Experience*, *Agreeableness* and *Conscientiousness* (and constituent features of these dimensions). The dimensions of the big five offer the possibility to comprehensively and factually describe, and interpret personality in terms of five major domains: general activity and contact, emotional balance, the way of relating to people, the way of relating to tasks, and general reference to the world and new experiences [14,16].

Personality is also important for the quality of food choices, which was demonstrated, among others, in young adults from New Zealand, university students from Ghana and physical education students from Poland and Spain [17,18,19]. Among physical education students, the intensity level of a pro-healthy diet has been described along with an increase in the levels of extraversion and conscientiousness [19]. In research on the subject, it has been shown that people practicing sports are characterised by a lower level of *Neuroticism*, which is conducive to more favourable eating behaviours [20]. In other studies, it has been indicated that *Conscientiousness* is associated with reduced exposure to psychoactive substances (alcohol and tobacco) and increased consumption of vegetables and fruits [21]. In research on the subject, it has also been demonstrated that high *Extraversion* and *Conscientiousness* were conducive to undertaking physical activity and applying a healthier lifestyle [22,23,24]. It was also confirmed that *Conscientiousness* and *Openness to Experience* are predictors of occupational burnout and lower quality of life among athletes [25]. A research hypothesis has been formulated in reference to the results of the above-mentioned research. The mostly positive predictive significance of extraversion and conscientiousness have been suggested, as well as the negative significance of neuroticism for health-related behaviours, including nutritional ones among various population groups [17,18,19,20,21]. Another premise of the adopted research hypothesis regarded the characteristics of individual personality dimensions, including neuroticism (associated with emotional lability), extraversion (associated with positive emotion) and conscientiousness (associated with the ability to control stimuli and focus on achievement striving) [14,15,16].

Due to the significance of diet quality for the health and exercise capacity of athletes, complexity regarding the determinants of nutritional behaviour, the lack of exploitation on the subject correlations between personality and diet as well as the ambiguity of the study results obtained so far, research was undertaken on the personality determinants regarding athletes’ diets. The aim of the study was to analyse the personality determinants of diet health quality among an elite group of Polish team athletes. The relationships between the personality traits of the big five model and the indices of diet health quality (pHDI-10 and nHDI-14) were assessed. The following research questions were posed: (1) What are the indices of diet health quality among athletes? (2) What are the personality traits of athletes? (3) What are the correlations between the personality traits and the diet quality among athletes? A research hypothesis was adopted stating that personality traits are related to the diet health quality of athletes, with higher indices of a pro-healthy diet fostered by *Conscientiousness* and *Extraversion*, and less favourably—by *Neuroticism*.

## 2. Materials and Methods

### 2.1. Participants

Research was carried out among a group of 213 Polish team athletes (males), including those practicing basketball (*n* = 54), volleyball (*n* = 53), football (*n* = 53) and handball (*n* = 53). The basic criterion for selection to the study group was practicing sports at a competitive level, at the level of the highest league in Poland, for at least 3 years. The study was conducted at 18 basketball clubs, 19 volleyball clubs, 13 soccer clubs and 10 handball clubs. The age of the tested athletes ranged from 18 to 38 years (M = 26.1; SD = 4.5), with sports experience being between 3 and 20 years (M = 8.2; SD = 4.5). The research was performed in accordance to the 1964 Declaration of Helsinki, after obtaining the participants’ written informed consent. The research protocol was approved by the Bioethics Committee at the Regional Medical Chamber in Kraków (No. 105/KBL/OIL/2021).

### 2.2. Instruments

#### 2.2.1. Assessment of Diet Health Quality

Diet was assessed using the beliefs and eating habits questionnaire created by the Behavioural Conditions of Nutrition Team, Committee of Human Nutrition Science, Polish Academy of Science [10]. This indicates a sufficiently high repeatability of results, which was confirmed in the validation procedure [26]. The frequency of consuming selected products was assessed on a 6-point ordinal scale, with the following ranks: 1 (never), 2 (1–3 times a month), 3 (once a week), 4 (several times a week), 5 (once a day), and 6 (several times a day). The original ranks were then converted into real numbers expressing the daily frequency of food consumption (as times/day), according to the formula: never (0), 1–3 times a month (0.06), once a week (0.14), several times a week (0.5), once a day (1) and several times a day (2) [10].

By summing up the values determining the daily frequency of consuming specific product groups, 2 indicators of diet quality were calculated: the pro-healthy diet index (pHDI-10), concerning the consumption of foods with potentially beneficial effects on health (10 groups of products) and the non-healthy diet index (nHDI-14), concerning the consumption of foods potentially detrimental to health (14 product groups). The indices are interpreted as follows: the higher the value of the index, the higher the intensity of the nutritional features favourable or unfavourable to health.

The values of the pHDI-10 index, expressed as the sum of the daily frequency of consuming 10 groups of products (times/day), are within the range of 0–20, while the values of the nHDI-14 are between 0–28. Values within the range of 0–6.66 for the pHDI-10 are interpreted as low, within the range of 6.67–13.33 as moderate, and between 13.34–20.0, as high. In the case of nHDI-14, values within the range of 0–9.33 are defined as low, 9.34–18.66 as moderate, and between 18.67–28.0, as high [10].

The pHDI-10 is related to the frequency consuming 10 groups of products: wholemeal bread, other whole grain cereal products, milk, fermented dairy drinks, cottage cheese, white meat dishes, fish, legumes, fruits and vegetables. The nHDI-14 is determined by the frequency of consuming 14 groups of products: light bread, other refined cereal products, fast food, fried foods, butter, lard, cheese and processed cheeses, meat products, red meat dishes, sweets, canned meat, sweetened carbonated or non-carbonated beverages, energy drinks, alcoholic beverages [10].

#### 2.2.2. Assessment of Personality Traits

To assess personality traits in the five-factor model, the NEO-PI-R (neuroticism extraversion openness personality inventory—revised) by P.T. Costa and R.R. McCrae was used [16] in the Polish adaptation [27]. The NEO-PI-R includes 240 statements, the truth of which is assessed by the respondent on a 5-point scale (from “completely disagree” to “completely agree”). The statements refer to 5 personality dimensions (scales), within which there are 6 subscales. The questionnaire includes: *Neuroticism* (anxiety, hostility/anger, depression, self-consciousness, impulsiveness/immoderation, vulnerability to stress/fear/learned helplessness), *Extraversion* (warmth/kindness, gregariousness/sociability, assertiveness, activity level/lively temperament, excitement seeking and positive emotion), *Openness to Experience* (fantasy/imagination, aesthetics/artistic interest, feelings/emotionality, action/adventurousness/exploration, ideas/intellectual interest/curiosity, values/psychological liberalism/tolerance to ambiguity), *Agreeableness* (trust in others, straightforwardness/morality, altruism, compliance/cooperation, modesty, tendermindedness/sympathy) and *Conscientiousness* (competence/self-efficacy, order(liness)/organising, dutifulness/sense of duty/obligation, achievement striving, self-discipline/willpower, deliberation/consciousness). Thus, the NEO-PI-R questionnaire allows for the description of 30 personality traits. The measurement reliability of the 5 dimensions in the Polish adaptation of the questionnaire is adequate and amounts to (respectively): *Neuroticism*—0.86, *Extraversion*—0.85, *Openness to Experience*—0.86, *Agreeableness*—0.81, *Conscientiousness*—0.85 [27]. In the case of subscales, the majority of reliability coefficients range from 0.60 to 0.70, while 8 subscales have reliability coefficients ranging from 0.51 to 0.57, and 2 below 0.50 (U6—*tendermindedness/sympathy*, and S4—*achievement striving*). In the case of these 2 subscales, the obtained results should be approached with caution [27].

### 2.3. Statistical Analysis

The collected numerical material was subjected to statistical analysis using the Statistica 13.3 package. Basic statistical measures (M—arithmetic mean, Me—median, SD—standard deviation, Q25—lower quartile, Q75—upper quartile, minimum and maximum) were calculated. Statistical analysis was performed using Pearson’s linear and Spearman’s signed rank correlation coefficients (depending on the nature of the variables). Multiple regression analysis was also implemented to identify which of the variables could explain the level diet health quality indices. In the calculations, a progressive stepwise regression procedure without constant terms was applied. The analysis also included calculation of multivariate determination coefficient (R^2^) values and the standard error of estimate (S_e_), as well as the values of standardised partial regression coefficients b*, which are a measure of relative significance concerning independent variables (i.e., personality traits) in the model. The analyses were performed with the significance level of α = 0.05.

## 3. Results

### 3.1. Health Quality of Athletes’ Diet

In Table 1, the frequency of consuming products with potentially beneficial and potentially adverse health effects among athletes is presented. Among the products with potentially beneficial effects on health, athletes ate fruits (49.3%) and vegetables (29.6%) in the highest proportion, at least once a day. However, the largest percentage of the respondents consumed fruits and vegetables several times a week, and whole grain cereal products, legumes, dairy products, poultry and fish dishes once a week. Among the products with potentially detrimental effects, the athletes consumed canned meats (53.5%), light bread (31.9%) and sweets as well as confectionery (31.9%) at least once a day. However, the highest percentage of subjects consumed light bread, sweets and confectionery several times a week. Other light cereal products, cheeses, red meat dishes, butter, sweetened carbonated and non-carbonated beverages and energy drinks were usually consumed once a week. Other products were eaten less frequently (Table 1).

In Table 2, the daily frequency of consuming products with potentially beneficial and potentially unfavourable effects on health is shown, as well as the values for pro-healthy (pHDI-10) and non-healthy diet (nHDI-14) indices. Among the products potentially beneficial to health, the studied athletes consumed fruits and vegetables the most (Me = 0.5), while the least frequently, fermented dairy products (Me = 0.06). Among the products with potentially adverse health effects, they most often consumed processed/canned meats (Me = 1.0), light bread, butter, sweets and confectionery (Me = 0.5), and least often, canned meats, fried dishes, fast food and alcoholic beverages (Me = 0.06). The value of the pro-healthy diet index (pHDI-10) was 3.83 times/day, while for its non-healthy counterpart, the level totalled 4.39 times/day (and, respectively: 19.16 and 15.69 points) (Table 2).

### 3.2. Athletes’ Personality Traits

Among the personality dimensions of the five-factor model, the surveyed athletes obtained the highest results in terms of *Conscientiousness* (M = 128.5), *Agreeableness* (M = 123.2), *Extraversion* (M = 121.8) and *Openness to Experience* (M = 115.0), and the lowest in terms of the level of *Neuroticism* (M = 72.15) (Table 3).

The levels of individual traits (subscales) regarding the five personality dimensions are presented in Table 4. Within the *Neuroticism* dimension, the athletes obtained the highest scores for *vulnerability to stress/fear/learned helplessness* (14.31) and *hostility/anger* (13.06). In terms of *Extraversion*, the highest results were related to *gregariousness/sociability* (23.54), and in terms of *openness to experience*—*values/psychological liberalism/tolerance to ambiguity* (21.05). Among the features of the conscientiousness dimension, the highest scores were obtained in terms of *straightforwardness/morality* (22.97), and the lowest in terms of *compliance/cooperation* (18.00), while among the features of the conscientiousness dimension, high scores were obtained for *duty/obligation* (24.71), and low for *deliberation/consciousness* (16.47) (Table 4).

### 3.3. Correlations between Personality Traits and Diet Health Quality of Athletes

In Table 5, correlations are presented between personality traits and diet health quality indices, and between personality traits as well as the daily frequency of consuming products with potentially beneficial and potentially unfavourable effects on health. It has been shown that the pro-healthy diet index (pHDI-10) decreased with increasing *Neuroticism* (r = −0.17), while it experienced a rise with increasing *Extraversion* (r = 0.13). Furthermore, the non-healthy diet index (nHDI-14) decreased with an increase in the level of *Agreeableness* (r = −0.17) (*p* < 0.05). It was shown that among the products with potentially beneficial effects on health, athletes consumed milk (R = 0.16) and vegetables (R = 0.15) more frequently as *Extraversion* increased, while they consumed sea fish (R = −0.14) and vegetables (R = −0.20) less often with increasing *Neuroticism*. Along with the increase in openness, the participants consumed legumes less frequently (R = −0.17); with increasing *Agreeableness*, they ate vegetables less often (R = −0.17), and with increasing *Conscientiousness*, they consumed vegetables at a higher frequency (R = 0.16) (*p* < 0.05). It has also been demonstrated that among products with potentially adverse health effects, with increasing O*penness to Experience*, the frequency of consuming refined cereal products (R = −0.17) and fried foods (R = −0.17) decreased as *Extraversion* increased. With the rise in *Extraversion*, the intake of sweets and confectionery products (R = 0.16) intensified, and with increasing *Agreeableness*, the frequency of drinking alcoholic beverages also increased (R = 0.16) (*p* < 0.05).

The analysis of correlations between the characteristics (subscales) of individual personality dimensions and the health quality of the athletes’ diet is presented in Table 6. It has been shown that the pro-healthy diet index (pHDI-10) increased with the decrease of several *Neuroticism* traits, including *anxiety* (R = −0.16), *hostility/anger* (R = −0.14), *depression* (R = −0.20) and self-*consciousness* (R = −0.15), as well as with the increase in the level of one of the features of *Extraversion*, i.e., *activity level/lively temperament* (R = 0.14). The pHDI-10 index also increased with the decrease in the level of one of the *Agreeableness* features, i.e., *modesty* (R = −0.17). In turn, the non-healthy diet index (nHDI-14) decreased with an increase in the level of one of the features of *Openness to Experience*, i.e., *fantasy/imagination* (R = −0.14), and with an increase in the level of one of the features of *Agreeableness*, i.e., *trust—in others* (R = −0.14) (*p* < 0.05). No significant correlations were found between the features (subscales) of *Conscientiousness* and the indices of diet health quality among the studied athletes (*p* > 0.05) (Table 6).

Multiple regression analysis (dependent variables: pHDI-10 and nHDI-14, predictors: personality traits of the big five model) allowed to show that the full model consisting of all analysed personality traits explains 91.8% of the variance for the pHDI-10 index, with significant predictors being *Extraversion* (positive correlation) and *Neuroticism* (negative correlation). This model also explains 88.2% of the variance in the level of the nHDI-14 index, with significant predictors being *Extraversion*, *Agreeableness* and *Neuroticism* (positive correlations) (Table 7).

## 4. Discussion

The research discussed in this paper allowed us to demonstrate a low level of diet health quality, a low level of neuroticism and significant correlations between certain personality dimensions and health quality indices of the diet among Polish team athletes (males).

When discussing the results in terms of diet health quality, referring to the applied methodology [10], a low level of the pro-healthy (19.16 points) and non-healthy diets (15.69 points) was shown. This should be interpreted as a limited, both positive and negative impact of diet on the health of the athletes under study. The low level of the pro-healthy diet index (pHDI-10) among athletes was related to the low frequency of consuming recommended products, including fruits and vegetables, wholemeal bread and other low-milled cereal products and legumes, as well as milk, dairy products and sea fish. These are products with high nutritional density, containing, among others, antioxidant substances, dietary fibre and omega-3 polyunsaturated acids, as well as probiotics, ingredients with significant pro-health values, including active ones, e.g., in the prevention of cardiovascular, metabolic and neoplastic diseases [28,29,30,31]. Due to the increased formation of free radicals under conditions of vigorous exercise, a diet rich in vegetables, fruits, legumes and specific products containing antioxidant ingredients (including polyphenols and vitamins C, E and carotenoids) is important for restoring pro-oxidative-antioxidant balance and reducing oxidative stress indices in athletes [31,32,33,34].

The demonstrated low level of the pro-healthy diet among elite Polish team athletes corresponds to nutritional, quantitative and qualitative irregularities shown in various groups of athletes, including those training team disciplines, both in Poland [35,36] and in other countries [37,38,39,40,41,42,43]. Varying levels of diet quality indices among athletes have also been found in various studies. High health quality of the diet (based on the athlete diet index, ADI), related to the sufficient consumption of recommended products, including vegetables, cereals and dairy, was described among Australian athletes of individual and team sports [44]. On the other hand, in Brazilian studies (using the healthy eating index, HEI), a low health quality of the diet was confirmed in approximately 73% of the group of athletes training volleyball, indicating a low supply of vitamins A and E [45]. An average level of diet quality (according to the HEI index) was described among athletes practicing team [46] and individual sports [47], and among academic athletes [48]. In Poland, the low health quality of diet (based on HDI indices) has also been described among young football players [49] and in other groups of individuals undertaking increased levels of physical activity, including physical education students [19,50] and aging masters athletes [51].

In our research, it was further shown that in terms of personality profile, the elite group of Polish team athletes was characterised by the highest scores regarding dimensions of conscientiousness, agreeableness, extraversion and openness, and the lowest in terms of neuroticism. Comparing raw scores to sten norms (for men aged 17–29 and 30–79) [27], it can be indicated that the studied athletes were characterised by a high level of conscientiousness, agreeableness, extraversion and openness (7–8 sten, depending on age) and a low level of neuroticism (3–4 sten, depending on age).

In our study, it was also shown that the elite group of Polish team athletes was characterised by a high level of the *Conscientiousness*, *Agreeableness* and *Extraversion* dimensions, a lower level of *Openness to Experience*, while the lowest level was *Neuroticism* in terms of the personality profile. It should also be pointed out that it is not the intention of the authors to conduct an advanced analysis of the personality profile of athletes, but only to comprise a general characteristic of the personality traits’ configuration to such an extent that it constitutes a background for the assessment of psychological determinants regarding diet health quality.

In further studies on the personality of athletes, different trends were shown. Athletes practicing team sports (football, handball and water polo) obtained the highest scores in terms of *Openness to Experience*, *Agreeableness* and *Extraversion*, but lower values in terms of *Conscientiousness*, and the lowest with regard to *Neuroticism* [52]. Similarly, in the next group of athletes performing team sports, the highest levels of *Openness to Experience* and the lowest of *Neuroticism* were demonstrated [53]. Therefore, it may be concluded that, despite the different configuration of features, the trend occurring in the majority of studies conducted among professional athletes is the lowest intensity of the *Neuroticism* dimension among the features included in the Costa and McCrae five-factor model. The low level of *Neuroticism* among athletes was also confirmed among Polish athletes by other authors [20], especially with regard to championship athletes [54,55].

Furthermore, statistically significant relationships were exhibited in our study between the personality traits of the big five model (and their subscales) and the assessed indices of diet health quality (however, mostly weak). In terms of the diet health quality indicators, predictive significance of *Neuroticism*, *Extraversion* and *Agreeableness* was found. The performed multiple regression analysis confirmed a very high share of the analysed personality traits in explaining the variance of diet quality indicators (above 88%). High *Neuroticism* (and its subscales: high *anxiety*, *hostility/anger*, *depression* and *self*-*consciousness*) was associated with a decrease in the health quality of the diet (i.e., a decrease in the pHDI-10 pro-healthy diet index) and a decrease in the frequency of consuming products with high nutritional density (e.g., vegetables and sea fish). High *Extraversion* (and its subscale: high *activity level/lively temperament*) was associated with an increase in the overall pro-healthy diet index (pHDI-10) and with a greater frequency of consuming products having high nutritional value (including vegetables and milk). High *Agreeableness* (and its subscales: high *fantasy/imagination* and high *trust in others*) was associated with a lower non-healthy diet index (nHDI-14), while high *modesty* was connected with a lower level of the pro-healthy diet index. On the other hand, significant correlations were found between some personality traits and the frequency of consuming individual products with potentially beneficial and potentially unfavourable effects on health, while unambiguous positive trends were shown only in terms of *Conscientiousness* (with its intensification, the consumption of vegetables increased). Other relationships were less clear.

The obtained results confirm the difficulties in unambiguous assessment of the relationship between personality traits and diet, including its health quality. The greatest number of unambiguous correlations at the level of overall nutritional indices was noted for *Neuroticism* (indicating its negative predictive importance for the health quality of diet) and *Extraversion* (indicating its positive predictive importance for the health quality of the diet). One can also indicate the dimension of *Openness to Experience* (with an indication of the positive meaning of one of the *fantasy/imagination* subscales).

Similar correlations between personality traits and food choices among various population groups were also described at others research centres, including in our previous studies. In analogous trials carried out among academic youth performing increased levels physical activity (Polish and Spanish physical education students), partially similar regularities were shown. It was noted among PE students that with the intensification of *Extraversion*, the level of both diet health quality indices (pHDI-10 and nHDI-14) increased (in Polish team athletes, only pHDI-10). At the same time, with increasing *Openness to Experience* and *Agreeableness*, the value of the non-healthy diet index (nHDI-14) decreased (in Polish team athletes, this relationship occurred only for *Agreeableness*). Along with the increase in the level of *Conscientiousness*, the pro-health quality of the diet increased (pHDI-10 index), which was not noted among the athletes under study [19]. At the level of personality relationships with the consumption of products recommended and not recommended in the diet, high correlations were noted for *Extraversion* and *Conscientiousness* with the consumption of fruit and vegetables, high *Extraversion* with the consumption of sweets and confectionery, and high *Neuroticism* with a low consumption of sea fish [19]. The indicated tendencies are consistent with the regularities described among the evaluated athletes performing team sports. The significance of personality traits for the quality of food choices was also described among academic youth from New Zealand (here, a positive correlation was found between *Extraversion* and fruit consumption) [17] and among students from Ghana (here, inter alia, between *Extraversion* and interest in new food products, associations of *Agreeableness* with irregular consumption of meals and *Conscientiousness* with a variety of diets and limiting sugar consumption) [18]. In contrast to our study, in research carried out among the Indonesian population, it has been shown that *Conscientiousness* is the only personality dimension that is clearly positively related to rational food choices [56]. Nonetheless, conclusions from other studies correspond to the results obtained in our research under discussion. Other studies conducted in Indonesia aimed at assessing the relationship between personality traits and body mass index (BMI), and demonstrated that individuals with excess body mass were less extroverted (i.e., more introverted) than people with normal body mass, which may indirectly indicate more rational food choices (and greater physical activity) of more extroverted people [57]. Research on the relationships of personality traits, eating behaviours and the BMI factor was also undertaken in the Australian population, which allowed the confirmation (among other findings) that nutrition, also applied in the Australian population, was considered a healthy pattern based on plant foods (vegetables, fruits, legumes) and fish was positively associated with *Openness*, *Conscientiousness* and *emotional stability*, and that BMI was negatively correlated with *Conscientiousness* and *emotional stability*, and positively with *Agreeableness* [58]. The cited Australian studies correspond with the results obtained for Polish team athletes with regard to correlations between low neuroticism levels and higher diet quality.

It may be summarised that various studies on personality determinants of nutritional behaviours in different population groups sometimes yield varied and ambiguous results. Nonetheless, the obtained results can be of practical usage for strategies aimed at changing (nutritional) behaviour due to the potential possibility of an interaction consistent with personality traits, assuming that personality is a rather permanent configuration of traits. However, further research conducted with interdisciplinary teams is needed to explain the mechanisms of the observed dependencies, which has also been pointed out by other authors [56].

Nutritional irregularities demonstrated among athletes justify the need to monitor diet and nutritional education, taking individualisation of effects promoting a healthy diet into account. Assessing the predictive significance of personality traits included in the big five model, in accordance with the sports nutrition model, should contribute to the effective rationalisation of diet through the individualisation of potential educational and dietary interactions due to the possibility of developing a strategy cohesive with personality profile.

The limitations of the work are primarily related to the lack of including demographic and sports variables (age, professional experience, discipline) in the analysed variables, taking one selected nutritional area (diet quality) into account and the self-report nature of the research tools used. It should also be highlighted that the achieved results only relate to men. The indicated and other limitations may determine the directions of further research, the aim and subject of which should be a comprehensive assessment of personality determinants regarding various areas of sports nutrition, taking age, gender, sports level and type of discipline into account. Further research in this area may include personality determinants of nutritional behaviours, including post-exercise nutrition and the use of dietary supplements (including ergogenic aids) by athletes. An important field in future research could also be the assessment of potential relationships between personality profile and quantitative aspects of sports nutrition (i.e., with the supply of energy and nutrients).

## 5. Conclusions

1. Among Polish team athletes, low levels of pro-healthy (pHDI-10) and unhealthy (nHDI-14) diet indices have been demonstrated, which means a low, both protective and negative impact of the diet on health.

2. High-class athletes training team sports are characterised by a high level of *Conscientiousness*, *Agreeableness, Extraversion* and *Openness* with a low level of *Neuroticism*.

3. Significant correlations between personality traits according to the big five model and the health quality of the Polish team athletes’ (males) diets were demonstrated, while the pro-healthy diet index (pHDI-10) decreased with increasing *Neuroticism*, and increased with increasing *Extraversion*. At the same time, the non-healthy diet index (nHDI-14) decreased as the level of *Agreeableness* increased; however, the strength of the correlations was low. At the same time, the greatest number of features negatively related to diet health quality was described in terms of the *Neuroticism* dimension (high level of *anxiety*, *hostility/anger*, *depression* and *self-consciousness*). Thus, the predictive significance of personality traits for the quality of sports nutrition choices was confirmed.

4. The obtained results indicate the validity of dietary monitoring and nutritional education in rationalising the diet and improving the health quality of (male) Polish team athletes’ diets. The correlations between personality traits and the quality of diet are not fully unambiguous and require further research so that they could serve the effective individualisation of educational and dietary interactions among the Polish team athletes.

## Figures and Tables

**Table 1 ijerph-19-16598-t001:** Frequency of consuming products with potentially beneficial (pHDI-10) and potentially detrimental (nHDI-14) impact on health among the surveyed athletes (N = 213) (percentage of respondents).

Indices/Food Products		Consumption Frequency					
		1	2	3	4	5	6
		% of Study Participants					
Potentially beneficial to health (pHDI-10)	Fruits	0.0	0.5	13.6	36.6	34.3	15.0
	Vegetables	0.0	4.2	24.9	41.3	26.3	3.3
	Wholemeal bread	12.7	30.5	31.9	18.8	5.6	0.5
	Coarse grains, oatmeal, whole grain pasta	2.8	13.1	43.2	40.4	0.5	0.0
	Legumes	6.1	35.7	37.6	19.2	1.4	0.0
	Milk	6.6	20.2	26.3	25.4	14.1	7.5
	Fermented dairy products	14.6	35.7	30.0	16.9	2.8	0.0
	Fromage frais	1.9	17.8	35.7	28.2	15.0	1.4
	Dishes from poultry	3.8	13.1	35.7	29.1	18.3	0.0
	Dishes from fish	1.4	24.9	34.7	26.3	12.7	0.0
Potentially detrimental to (nHDI-14)	Light bread	8.0	14.1	15.5	30.5	15.5	16.4
	White rice, fine groats	1.4	25.4	38.5	34.3	0.0	0.5
	Yellow, blue and processed cheeses	4.2	34.7	39.4	16.4	4.7	0.5
	Cold-cuts, sausages, hot dogs	1.4	1.4	15.0	28.6	30.5	23.0
	Dishes from red meat	2.8	18.8	40.8	27.7	9.4	0.5
	Processed meats	12.2	47.9	29.1	8.9	1.9	0.0
	Fried dishes	17.8	52.6	14.1	15.5	0.0	0.0
	Butter	14.1	8.9	26.8	22.5	16.9	10.8
	Lard	91.1	7.0	1.9	0.0	0.0	0.0
	Fast food	33.8	43.2	23.0	0.0	0.0	0.0
	Sweets and confectionery	1.4	7.0	27.2	32.4	19.2	12.7
	Sweetened carbonated and non-carbonated drinks	8.9	21.6	47.4	15.0	7.0	0.0
	Energy drinks	12.2	26.3	39.4	17.4	4.7	0.0
	Alcoholic beverages	19.2	57.3	21.6	1.9	0.0	0.0

Legend: (1) Never (2) 1–3 times a month (3) Once a week (4) A few times a week (5) Once a day (6) A few times a day.

**Table 2 ijerph-19-16598-t002:** Daily frequency of consuming products with potentially beneficial and potentially detrimental impact on health and value of pro-healthy (pHDI-10) and non-healthy (nHDI-14) diet indices (N = 213) among the studied athletes (N = 213) (descriptive statistics).

Indices/Products		M	SD	Me	Min	Max	Q25	Q75
pHDI-10	Fruits	0.85	0.57	0.50	0.06	2.00	0.50	1.00
	Vegetables	0.57	0.42	0.50	0.06	2.00	0.14	1.00
	Wholemeal bread	0.22	0.28	0.14	0.00	2.00	0.06	0.14
	Coarse grains, oatmeal, whole grain pasta	0.27	0.20	0.14	0.00	1.00	0.14	0.50
	Legumes	0.18	0.19	0.14	0.00	1.00	0.06	0.14
	Milk	0.47	0.54	0.14	0.00	2.00	0.06	0.50
	Fermented dairy products	0.18	0.22	0.06	0.00	1.00	0.06	0.14
	Fromage frais	0.38	0.38	0.14	0.00	2.00	0.14	0.50
	Dishes from poultry	0.39	0.34	0.14	0.00	1.00	0.14	0.50
	Dishes from fish	0.32	0.31	0.14	0.00	1.00	0.06	0.50
nHDI-14	Light bread	0.67	0.67	0.50	0.00	2.00	0.14	1.00
	White rice, fine groats	0.25	0.23	0.14	0.00	2.00	0.06	0.50
	Yellow, blue and processed cheeses	0.21	0.26	0.14	0.00	2.00	0.06	0.14
	Cold-cuts, sausages, hot dogs	0.93	0.66	1.00	0.00	2.00	0.50	1.00
	Dishes from red meat	0.31	0.31	0.14	0.00	2.00	0.14	0.50
	Processed meats	0.13	0.18	0.06	0.00	1.00	0.06	0.14
	Fried dishes	0.13	0.16	0.06	0.00	0.50	0.06	0.14
	Butter	0.54	0.61	0.50	0.00	2.00	0.14	1.00
	Lard	0.01	0.02	0.00	0.00	0.14	0.00	0.00
	Fast food	0.06	0.05	0.06	0.00	0.14	0.00	0.06
	Sweets and confectionery	0.65	0.60	0.50	0.00	2.00	0.14	1.00
	Sweetened carbonated and non-carbonated drinks	0.22	0.26	0.14	0.00	1.00	0.06	0.14
	Energy drinks	0.20	0.24	0.14	0.00	1.00	0.06	0.14
	Alcoholic beverages	0.07	0.07	0.06	0.00	0.50	0.06	0.06
Indices	pHDI-10 (times/day)	3.83	1.16	3.70	1.58	7.20	3.02	4.48
	nHDI-14 (times/day)	4.39	1.59	4.14	1.28	9.40	3.30	5.32
	pHDI-10 (points)	19.16	5.80	18.50	7.90	36.00	15.10	22.40
	nHDI-14 (points)	15.69	5.67	14.79	4.57	33.57	11.79	19.00

Legend: M arithmetic mean, SD standard deviation, Me median, Q25 lower quartile, Q75 upper quartile.

**Table 3 ijerph-19-16598-t003:** Level of personality traits according to the five-factor model among athletes training team sports (N = 213) (descriptive statistics).

Personality Traits	M	SD	Min	Max	Q25	Me	Q75
Neuroticism	72.15	20.55	23.0	128.0	56.00	71.00	89.00
Extraversion	121.80	15.52	70.00	151.0	111.0	124.0	133.0
Openness	115.00	13.91	92.00	141.0	101.0	115.0	129.0
Agreeableness	123.20	13.14	86.00	146.0	118.0	126.0	132.0
Conscientiousness	128.50	22.22	83.00	168.0	111.0	133.0	144.0

Legend: M arithmetic mean, SD standard deviation, Me median, Q25 lower quartile, Q75 upper quartile.

**Table 4 ijerph-19-16598-t004:** The level of traits of individual personality dimensions in the five-factor model in the studied group of sportsmen training team games (N = 213) (descriptive statistics).

Neuroticism			Extraversion			Openness to Experience			Agreeableness			Conscientiousness		
	M	SD		M	SD		M	SD		M	SD		M	SD
N1	10.62	4.93	E1	23.54	2.62	O1	18.62	3.69	A1	21.94	2.56	C1	23.24	4.44
N2	13.06	4.19	E2	20.07	3.93	O2	17.10	3.13	A2	22.97	3.55	C2	20.53	5.30
N3	12.29	4.16	E3	17.67	4.25	O3	20.51	3.80	A3	22.71	3.08	C3	24.71	3.78
N4	12.97	4.27	E4	21.47	4.39	O4	18.16	3.04	A4	18.00	3.58	C4	21.89	4.39
N5	14.31	5.38	E5	16.53	5.12	O5	19.51	3.77	A5	18.86	4.70	C5	21.61	4.38
N6	8.91	3.30	E6	22.50	2.53	O6	21.05	2.78	A6	18.68	2.57	C6	16.47	3.72

Legend: M arithmetic mean, SD standard deviation; N1 anxiety, N2 hostility/anger, N3 depression, N4 impulsiveness/immoderation, N5 vulnerability to stress/fear/learned helplessness, N6 self-consciousness; E1 gregariousness/sociability, E2 warmth/kindness, E3 assertiveness, E4 activity level/lively temperament, E5 excitement seeking, E6 positive emotion; O1 fantasy/imagination, O2 aesthetics/artistic interest, O3 feelings/emotionality, O4 action/adventurousness/exploration, O5 ideas/intellectual interest/curiosity, O6 values/psychological liberalism/tolerance to ambiguity; A1 trust in others, A2 straightforwardness/morality, A3 altruism, A4 compliance/cooperation, A5 modesty, A6 tendermindedness/sympathy; C1 competence/self-efficacy, C2 order(liness)/organising, C3 dutifulness/sense of duty/obligation, C4 achievement striving, C5 self-discipline, C6 self-discipline/willpower, deliberation/consciousness.

**Table 5 ijerph-19-16598-t005:** Correlations between the personality traits of the five-factor model and the indices of pro-healthy (pHDI-10) and non-healthy (nHDI-14) diets, personality traits as well as the daily frequency of consuming products with potentially beneficial and potentially adverse effects on health (N = 213) (Pearson’s r and Spearman’s R signed rank correlation coefficients).

Indices/Products		N	E	O	A	C
		Pearson’s r				
Indices	pHDI−10	−0.17 *	0.13 *	−0.00	−0.06	0.02
	nHDI−14	0.01	0.11	−0.01	−0.17 *	−0.01
		**Spearman’s R**				
Products from the pHDI−10 group	Wholemeal bread	−0.08	0.08	0.00	−0.08	0.04
	Coarse grains, oatmeal, whole grain pasta	0.08	0.12	−0.00	0.01	−0.02
	Milk	−0.03	0.16 *	−0.04	−0.04	0.02
	Fermented dairy products	0.01	−0.14 *	0.04	0.05	−0.00
	Fromage frais	−0.03	0.13	−0.02	−0.10	−0.02
	Dishes from white meat	0.18 *	−0.03	−0.00	0.06	−0.07
	Fish, seafood	−0.14 *	−0.06	0.08	−0.09	0.10
	Legumes	−0.08	0.04	−0.17 *	−0.13	−0.04
	Fruits	−0.05	−0.00	−0.02	0.09	−0.00
	Vegetables	−0.20 *	0.15 *	0.11	−0.17 *	0.16 *
Products from the nHDI−14 group	Light bread	0.04	0.05	0.13	0.02	−0.03
	White rice, normal pasta, small groats	0.10	0.00	−0.17 *	−0.12	0.00
	Fast food	0.00	−0.09	0.00	0.13	−0.01
	Fried dishes (meat, flour−based)	0.06	−0.00	−0.17 *	0.05	−0.02
	Butter	0.09	−0.02	−0.04	0.09	0.02
	Lard	−0.08	−0.02	−0.07	0.00	0.07
	Yellow, blue and processed cheeses	−0.11	−0.05	0.09	−0.03	0.13
	Cold−cuts, sausages and hot dogs	0.06	0.07	−0.01	−0.13	−0.05
	Dishes from red meat	−0.01	0.02	−0.02	−0.10	−0.08
	Sweets and confectionery	0.00	0.16 *	−0.05	−0.09	−0.03
	Processed meats	0.00	−0.04	−0.10	−0.01	−0.00
	Sweetened carbonated and non−carbonated drinks	0.01	0.05	0.01	−0.11	−0.06
	Energy drinks	0.00	−0.03	−0.01	0.08	0.03
	Alcoholic beverages	0.04	−0.03	−0.10	0.16 *	−0.09

* *p* < 0.05; N Neuroticism, E Extraversion, O Openness to Experience, A Agreeableness, C Conscientiousness.

**Table 6 ijerph-19-16598-t006:** Correlations between the subscales of personality dimensions and indices of diet health quality among the studied athletes training team sports (N = 213) (Spearman’s rank coefficient values).

Main Dimensions of Personality	Traits—Subscales of Personality Dimensions	pHDI−10	nHDI−14
Neuroticism	N1—Anxiety	−0.16 *	−0.04
	N2—Hostility/Anger	−0.14 *	0.07
	N3—Depression	−0.20 *	−0.02
	N4—Impulsiveness/Immoderation	−0.11	−0.02
	N5—Vulnerability to stress/Fear/Learned helplessness	−0.09	0.13
	N6—Self−consciousness	−0.15 *	0.08
Extraversion	E1—Gregariousness/Sociability	0.04	−0.04
	E2—Warmth/Kindness	0.05	0.08
	E3—Assertiveness	0.12	0.06
	E4—Activity level/Lively temperament	0.14 *	−0.04
	E5—Excitement seeking	0.05	0.10
	E6—Positive emotion	0.07	0.05
Openness to Experience	O1—Fantasy/Imagination	−0.09	−0.14 *
	O2—Aesthetics/Artistic interest	0.01	−0.04
	O3—Feelings/Emotionality	0.09	0.02
	O4—Action/Adventurousness/Exploration	0.07	0.03
	O5—Ideas/Intellectual interest/Curiosity	−0.11	−0.10
	O6—Vales/Psychological liberalism/Tolerance to ambiguity	0.03	0.10
Agreeableness	A1—Trust—in others	0.05	−0.14 *
	A2—Straightforwardness/Morality	−0.04	−0.09
	A3—Altruism	−0.03	−0.11
	A4—Compliance/Cooperation	−0.08	−0.06
	A5—Modesty	−0.17 *	−0.10
	A6—Tendermindedness/Sympathy	0.01	−0.05
Conscientiousness	C1—Competence/Self−efficacy	0.07	−0.02
	C2—Order(lines)/Organising	0.02	−0.01
	C3—Dutifulness/Sense of duty/Obligation	0.04	−0.05
	C4—Achievement striving	0.07	0.03
	C5—Self−discipline/Willpower	0.03	−0.07
	C6—Deliberation/Consciousness	−0.04	0.03

* *p* < 0.05.

**Table 7 ijerph-19-16598-t007:** Correlations between indices of diet health quality (pHDI-10 and nHDI-14) and personality traits in the big five model (multiple regression analysis).

Variables		R^2^	s_e_	b*	Std. Error from b*	B	Std. Error from b
pHDI-10	Extraversion	0.918	1.155	0.608	0.101	0.020	0.003
	Agreeableness			0.497	0.118	0.016	0.004
	Neuroticism			−0.149	0.074	−0.008	0.004
nHDI-14	Extraversion	0.882	1.613	0.677	0.121	0.026	0.005
	Agreeableness			0.173	0.141	0.007	0.005
	Neuroticism			0.097	0.088	0.006	0.006

R^2^ multidimensional coefficient of determination; Standard error of estimation; b* standardised partial regression coefficient.

## Data Availability

Not applicable.

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
