# Peer review of "Personality Determinants of Diet Health Quality among an Elite Group of Polish Team Athletes"

_ijerph, 2022, doi:10.3390/ijerph192416598_

Round 1

Reviewer 1 Report

Thank you for the Author's contribution to this study. The manuscript has an important topic, and it is well structured as well. 

            The introduction starts with the highlight of the importance of diet in sports. However, I recommend expanding this part and adding more information on athletes’ diets. The associations between diet and psychology are well-detailed and the research questions and study goal is clear. 

            The methods and the result are extremally details. I only have one recommendation to the Authors: The instruments could be deviled into subsections (e.g., 2.2.1... etc.) since it will be more convenient to separate and easier for the readers. 

            The discussion and the conclusion were well written. Maybe future direction could be added to this part. 

Author Response

Dear Reviewer, 

please see the attachment,

kind regards, MG 

Reviewer 2 Report

Comments and suggestions are in the file.

Author Response

(The authors gave the same response as above.)

Reviewer 3 Report

I find the paper well-written and documented. It is clear and comprehensive. The cited references are mostly recent and relevant. The tables are clear and well-structured.

I would just like to know why women were excluded from the study? It needs to be underlined in the study "conclusions" section as well as in the first paragraph of the "discussion" section and "abstract" that the research (and its results) refer only to males and therefore cannot be applied to the whole population of Polish athletes.

Author Response

(The authors gave the same response as above.)

Round 2

Reviewer 2 Report

Thank you for considering and responding to all comments. Congratulations on a great job.
I would just like to draw attention to one more small detail - the lack of a symbol in line 189. Regarding table 3 - I used the wrong word. "Unreadable" does not reflect what I meant. I wanted to say that table 3 is mismatched with the others in terms of style. All tables have single lines and this one has double lines. Good luck with your further scientific work. Kind regards

Author Response

Dear Reviewer,

you very much for your greatly extensive review of the work and the effort put into its preparation. Thank you also for accepting the explanations and corrections and for the kind words about the work done. Thank you very much. At the same time, I kindly inform you that I have added the missing 'median' symbol and corrected the double lines in table 3. 

Yours sincerely, MG